# Lin28 and *let-7* regulate the timing of cessation of murine nephrogenesis

Alena V. Yermalovich[1,2,3], Jihan K. Osborne[1,2,3], Patricia Sousa[1,2,3], Areum Han[1,2,3], Melissa A. Kinney[1,2,3], Michael J. Chen [1,2,3], Daisy A. Robinton[1,2,3], Helen Montie[1,2,3], Dan S. Pearson[1,2,3], Sean B. Wilson [4], Alexander N. Combes[4,5], Melissa H. Little [4,6] & George Q. Daley[1,2,3]

In humans and in mice the formation of nephrons during embryonic development reaches completion near the end of gestation, after which no new nephrons are formed. The final nephron complement can vary 10-fold, with reduced nephron number predisposing individuals to hypertension, renal, and cardiovascular diseases in later life. While the heterochronic genes *lin28* and *let-7* are well-established regulators of developmental timing in invertebrates, their role in mammalian organogenesis is not fully understood. Here we report that the Lin28b/*let-7* axis controls the duration of kidney development in mice. Suppression of *let-7* miRNAs, directly or via the transient overexpression of *LIN28B*, can prolong nephrogenesis and enhance kidney function potentially via upregulation of the *Igf2/H19* locus. In contrast, kidney-specific loss of *Lin28b* impairs renal development. Our study reveals mechanisms regulating persistence of nephrogenic mesenchyme and provides a rationale for therapies aimed at increasing nephron mass.

[1] Division of Pediatric Hematology/Oncology, Children's Hospital Boston, Boston, MA 02115, USA. [2] Department of Biological Chemistry and Molecular Pharmacology, Harvard Medical School, Boston, MA 02115, USA. [3] Harvard Stem Cell Institute, Boston, MA 02115, USA. [4] Murdoch Childrens Research Institute, Royal Children's Hospital, 50 Flemington Rd, Parkville, Melbourne 3052, Australia. [5] Department of Anatomy and Neuroscience, Faculty of Science, Building 181, Corner of Grattan Street and Royal Parade, University of Melbourne, Parkville 3010, Australia. [6] Department of Pediatrics, The University of Melbourne, Level 2 West, The Royal Children's Hospital, 50 Flemington Road, Parkville 3010, Australia. These authors contributed equally: Alena V. Yermalovich, Jihan K. Osborne.  Correspondence and requests for materials should be addressed to M.H.L. (email: Melissa.Little@mcri.edu.au) or to G.Q.D. (email: George_Daley@hms.harvard.edu)

Among its many functions, the mammalian kidney removes nitrogenous waste, regulates blood volume, and maintains bone density. Highly specialized epithelial tubules called nephrons serve as the basic functional units of the kidney[1]. Kidney development, or nephrogenesis, is a complex process that requires reciprocal inductive interactions between two precursor tissues derived from the intermediate mesoderm (IM): the metanephric mesenchyme (MM) and the ureteric bud (UB). The UB gives rise to the branching epithelium of the collecting ducts while the MM gives rise to the cap mesenchyme (CM) as well as stromal populations[2–5]. In mouse, the CM has been shown to represent a pool of multipotent nephron progenitors which self-renew and give rise to mature nephrons via a mesenchymal-to-epithelial transition (MET)[6,7]. Nephron formation continues within an outer nephrogenic zone of the kidney until postnatal day 2 in mice[8], and the 36th week of gestation in humans[9], after which time all remaining nephron progenitors undergo a synchronous wave of differentiation to establish the final number of nephrons—the "nephron endowment"—that will persist lifelong in the adult[8]. A human kidney contains anywhere from 200,000 to over 1.8 million nephrons[10]. Children who are born prematurely or suffer from intrauterine growth restriction (IUGR) as a result of maternal malnutrition have a reduced number of nephrons, which negatively affects the filtration function of the kidney. Because new nephrons do not form in the extra-uterine environment, children with a compromised nephron endowment are at increased risk of hypertension and development of cardiovascular and renal diseases, as well as insulin resistance and type 2 diabetes as adults[11–13]. Therefore, there has been interest in developing approaches to the treatment and prevention of kidney disease.

The RNA-binding protein Lin28 and the *let-7* microRNA (miRNA) family were originally discovered in *Caenorhabditis elegans* as heterochronic genes regulating developmental timing[14,15]. In mammals, *Lin28a* and its paralog *Lin28b* are highly expressed in stem and progenitor cells, where they function to inhibit biogenesis of the *let-7* family of miRNAs. As progenitor cells differentiate, Lin28 expression decreases, allowing formation of mature *let-7* miRNAs[16,17]. Members of the *let-7* miRNA family, in turn, bind to the 3′ UTR of *Lin28* mRNA, negatively regulating its expression. Thus, Lin28 proteins and *let-7* miRNAs mutually suppress each other to form a bistable switch that is conserved throughout evolution from worms to mammals[18–20]. Lin28 proteins also bind mRNAs and modulate their translation independently of modulation of *let-7*[21–25]. Aside from their role as developmental regulators, *Lin28a/b* and *let-7* genes have been implicated in metabolism[26], wound healing[27], and oncogenesis[18,28,29]. We have recently reported that prolonged expression of *Lin28* in developing kidneys in mice markedly expands nephrogenic progenitors, blocks their final wave of differentiation, and ultimately results in neoplastic transformation resembling the most common renal neoplasm of childhood, Wilms tumor, via misregulation of *let-7* miRNAs[30]. Wilms tumor shares histological features with the developing kidney, and arises from inappropriately persisting MM, providing a window into the mechanisms of early renal development and into the properties of embryonic kidney stem cells[31].

Given that *Lin28* and *let-7* genes were initially identified as heterochronic genes, we hypothesized that the Lin28/*let-7* axis may control cessation of nephrogenesis and hence perturbation to these genes might prolong nephrogenesis and increase nephron endowment. In this study, we show an inverse temporal pattern of expression for *Lin28a/b* and *let-7* transcripts during kidney development implicating Lin28b (and not Lin28a) as playing the predominant role in nephrogenesis. A single pulse of *LIN28B* overexpression in the Wt1-expressing MM during kidney

development results in the formation of a postnatal ectopic nephron forming population leading to a substantial increase in organ volume, a two-fold increase in the final nephron number and a concomitant increase in the filtration function of the kidney. Suppression of *let-7* miRNAs during kidney development also prolongs nephrogenesis, in this instance restricted to the peripheral nephrogenic zone, and enhances kidney function. Conversely, kidney-specific loss of *Lin28b* impairs renal development and function apparently via regulation of *let-7* miRNAs. Both overexpression of *LIN28B* and knockout (KO) of *let-7* miRNAs results in an upregulation of the *Igf2/H19* locus that has previously been associated with the persistence of metanephric blastema in both Wilms tumor and persistent nephrogenic rests persistence of a nephrogenic mesenchyme[32]. These data provide a rationale for manipulating the Lin28/*let-7* pathway to prolong nephrogenesis and suggest that this approach might enable a rescue of low nephron endowment and restoration of kidney function.

## Results

**Lin28/*let-7* expression during kidney development**. Our previous studies enforced overexpression of *Lin28a* or *LIN28B* from a variety of kidney specific promoters, including *Foxd1*, *Six2* and *Cdh16*, with no tumor formation, however, overexpression of *LIN28B* driven via a *Wt1* promoter surprisingly showed overt expansion of a Six2-expressing blastema, reminiscent of Wilms tumor[33]. This prompted us to explore the expression dynamics of *Lin28* during normal nephrogenesis in the mouse embryonic kidney. Western blot analysis reveals that while both Lin28a and Lin28b proteins are expressed at high levels in midgestation, *Lin28a* expression decreases at embryonic day 13.5 (E13.5), whereas *Lin28b* expression is prolonged but exhibits a rapid decline after E16.5 (Fig. 1a), coinciding with the functional maturation of the first nephrons in the developing kidney[31]. Quantitative reverse transcription polymerase chain reaction (qRT-PCR) shows that the amount of *Lin28b* mRNA is significantly higher relative to *Lin28a* at all time points tested (Fig. 1b). Along with previous data showing the prevalence of LIN28B (but not LIN28A) activation in human Wilms tumor[30], this strongly suggests that *Lin28b* plays the predominant role in normal kidney development.

To understand the role of *let-7* miRNAs in the developing kidney, we analyzed mature and precursor *let-7* miRNA levels in wild-type embryonic kidneys by qRT-PCR. All eight mature *let-7* family members follow a similar pattern: very low levels of expression up until E14.5 when both Lin28 paralogs are present. At E14.5, when Lin28a expression ceases, mature *let-7* miRNAs start to increase. However, the most significant change in the mature *let-7* miRNAs is observed between E14.5 and E16.5, which correlates with the gradual decrease of Lin28b expression (Fig. 1c; Supplementary Figure 1a). To determine whether *let-7* miRNAs are suppressed during early nephrogenesis due to the presence of Lin28a and/or Lin28b, we measured expression of the *let-7* precursor miRNAs. Interestingly, while one group of precursors, *pre-let7-a2*, *pre-let7-b*, *pre-let7-c1*, *pre-let7-d*, *pre-let7-e*, has the same pattern of expression as their mature family members (Fig. 1c, d; Supplementary Figure 1a, b), the second group, *pre-let7-a1*, *pre-let7-c2*, *pre-let7-f1*, *pre-let7-f2*, *pre-let7-g1*, and *pre-let7-i*, is upregulated in early-mid nephrogenesis (E12.5–E16.5) (Fig. 1c, d; Supplementary Figure 1a, b). These data suggest that Lin28b likely contributes to early kidney development by suppression of the latter group of precursor miRNAs.

**Prolonged expression of *LIN28B* increases nephron number.** We have previously shown that continuous induction of *LIN28B*

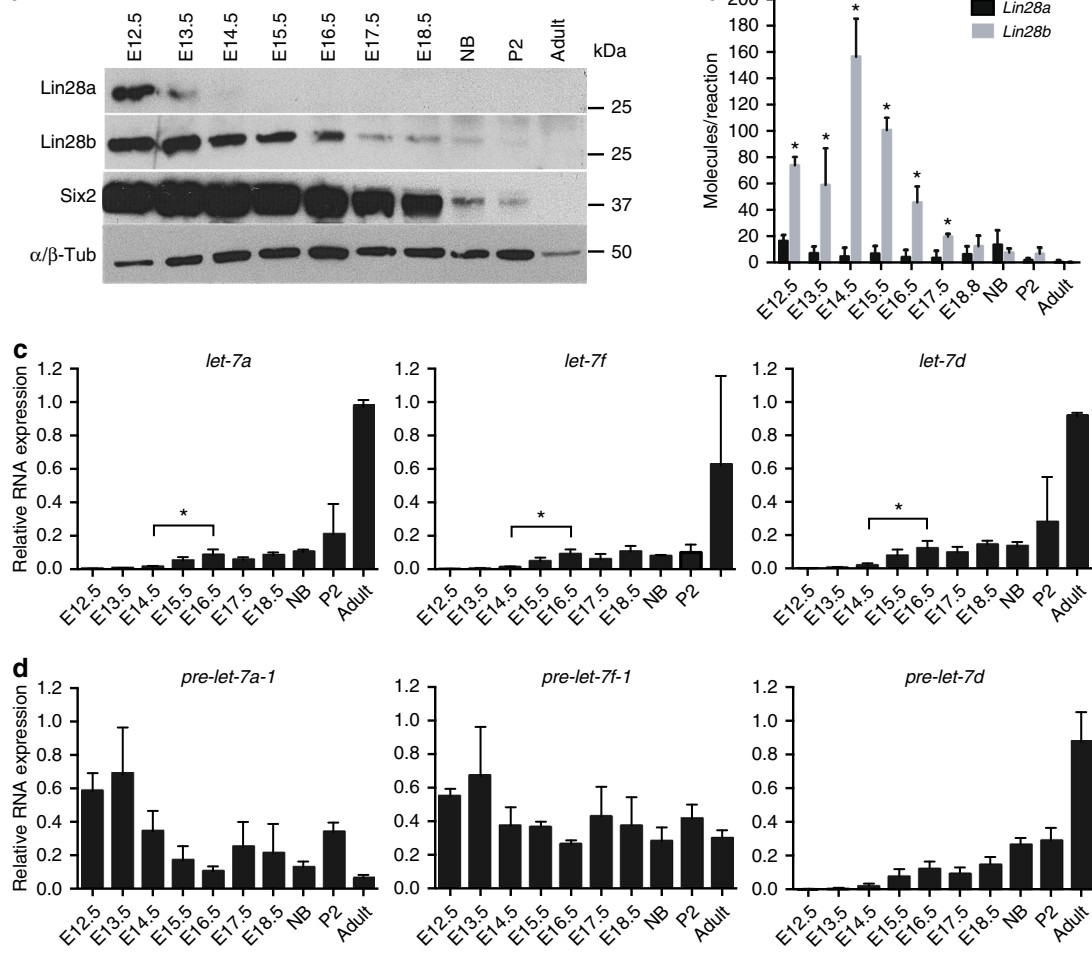

**Fig. 1** Expression of *Lin28* and *let-7* in mouse embryonic kidney. **a** Western blot analysis against Lin28a, Lin28b, Six2, and α/β-Tubulin proteins in lysates collected from dissected embryonic day 12.5–18.5 (E12.5–E18.5), newborn (NB), postnatal day 2 (P2), and adult wild-type kidneys. **b** Absolute qRT-PCR analysis measuring the levels of *Lin28a* and *Lin28b* mRNAs in wild-type kidneys. $n = 3$ for each sample type. **c**, **d** Relative qRT–PCR analysis measuring the levels of mature and precursor (pre-) *let-7* miRNAs in wild-type kidneys. $n = 3$–5 for each sample type. Error bars represent mean ± SD. (*) = $p < 0.05$, unpaired, two-tailed Student's *t* test

during embryonic development sustains and expands MM and/or nephron progenitors and blocks the final wave of nephrogenesis, ultimately resulting in oncogenic transformation resembling Wilms tumor. Withdrawal of *LIN28B* expression reverted tumorigenesis and CM cells eventually underwent terminal differentiation[30]. As *lin-28* was initially identified as a heterochronic gene in *C. elegans*[14,15], we hypothesized that *Lin28b* controlled the timing of cessation of nephrogenesis in mammals. To test this hypothesis, we utilized our previously generated gain-of-function *LIN28B* mouse model (*TRE-LIN28B; lox-STOP-lox-rtTA*) with a *Wt1-Cre* driver, enabling spatial and temporal control of the human LIN28B protein specifically in early kidney progenitors (henceforth referred to as *LIN28B^Wt1* mice)[30]. In this model system, *LIN28B* expression is controlled spatially by the expression of Cre in *Wt1*-expressing cells and temporally by administration of doxycycline (dox). *Wt1* is expressed in the IM, the earliest precursor of the complete metanephric kidney[30]. It is also expressed in the MM within the developing kidney, including both the cortical stroma and the CM, as well as the nephrons formed after induction of this mesenchyme, and ultimately in the podocytes of the glomeruli[34]. Notably, it was only overexpression of *LIN28B* driven by this promoter that previously resulted in tumor formation.

To examine whether we could subtly prolong nephron formation, *LIN28B* expression was induced using a single 1-day pulse of dox at E16.5, the time point at which endogenous expression of Lin28b protein declines rapidly during kidney development. The induction led to persistence of *LIN28B* mRNA and protein levels out to postnatal day 5 (P5) in *LIN28B^Wt1* kidneys (Fig. 2a, b), resulting in a consequent suppression of *let-7* miRNAs during this period due to high-level overexpression of ectopic *LIN28B* (Fig. 2c; Supplementary Figure 2a). We have analyzed the expression of endogenous *Lin28a* and *Lin28b* mRNA in *LIN28B^Wt1* kidneys and observe no change in transcript levels between *LIN28B^Wt1* and littermate controls at all the time points tested (Supplementary Figure 2b). This single-day induction resulted in the formation of ectopic fields of nephrogenic mesenchyme through to P14, as demonstrated by the persistent expression of CM-specific transcription factors Six2 and Eya1 (Fig. 2d–f; Supplementary Figure 2c). Indeed, blastema was evident even at P21 (Supplementary Figure 3) with such fields not only evident around the cortical nephrogenic zone but abutting collecting duct epithelium deep within the subcortical parenchyma. Immunohistochemistry staining for Lef1, a marker of the renal vesicles that represent the first epithelial derivatives of the CM, revealed that over time these ectopic Six2+ fields

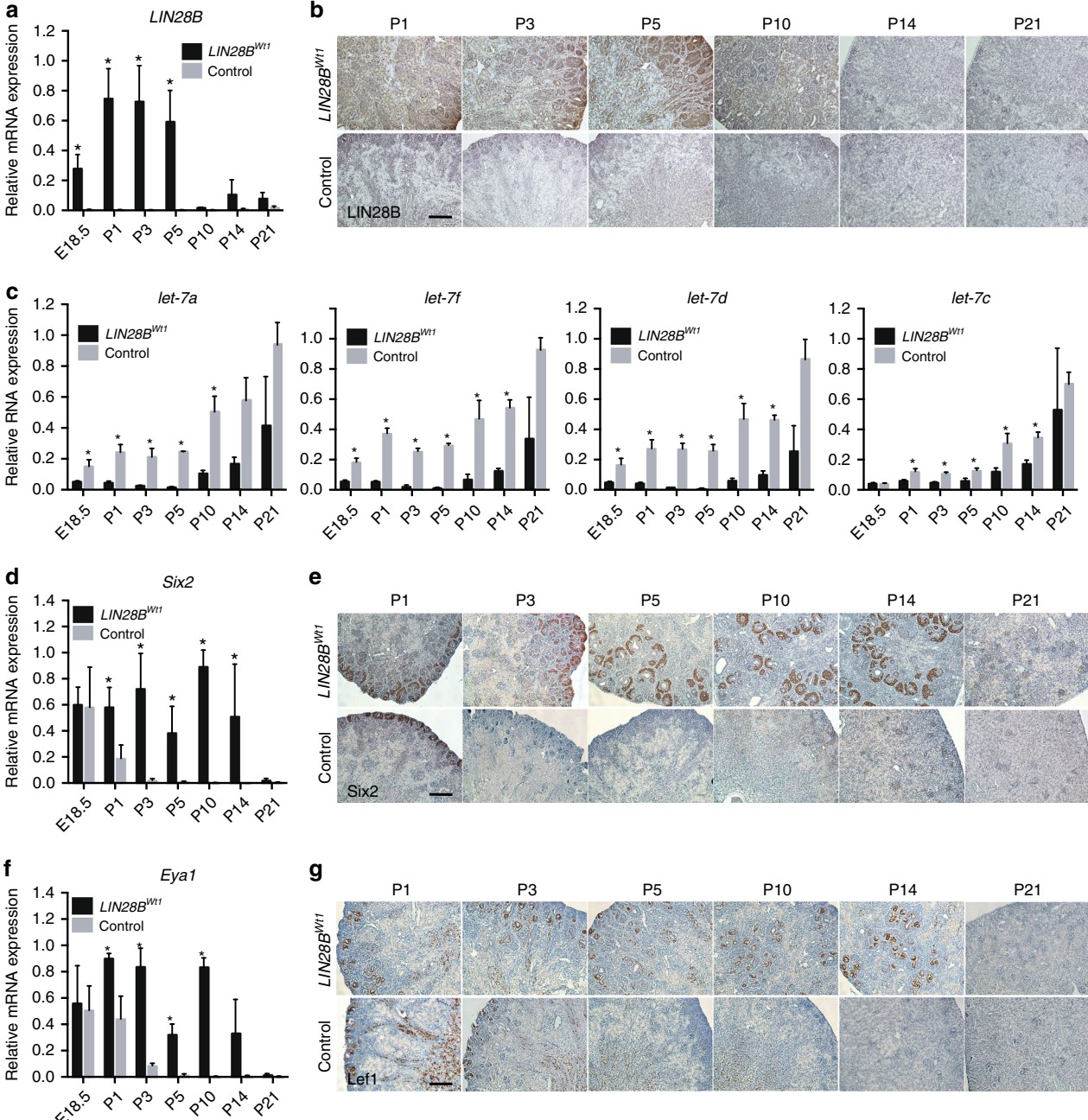

**Fig. 2** LIN28B overexpression in embryonic kidney enhances nephrogenesis. **a**, **c**, **d**, **f** Relative qRT–PCR analysis measuring levels of LIN28B, mature let-7, Six2, and Eya1 RNAs, respectively, in LIN28B^Wt1 and littermate control kidneys at the indicated developmental time points. n = 3 for each genotype. Error bars represent mean ± SD. (*) = p < 0.05, unpaired, two-tailed Student's t test. **b**, **e**, **g** Representative immunohistochemistry staining against LIN28B, Six2, and Lef1 proteins, respectively, in LIN28B^Wt1 and littermate control kidneys at the indicated developmental time points. Scale bar, 200 µm

underwent a mesenchymal to epithelial transition and subsequently form mature nephrons throughout the renal parenchyma (Fig. 2g).

The phenotype of the LIN28B^Wt1 kidneys is unique. There is little clear histological difference with control kidneys at birth, however, within the first few days of postnatal life, there is clear evidence for an expansion and mispositioning of the remaining CM with areas of CM extending deep into the parenchyma (Supplementary Figure 3). As this is not a location for the initial CM, it is likely that the enforced expression of LIN28B under Wt1-Cre promoter is resulting in induction of this gene in mesenchymal populations other than the CM, resulting in the formation of ectopic CM

domains surrounding and apparently initiating new lateral buds along the collecting ducts (Supplementary Figure 3). This provides evidence showing that the postnatal collecting duct can respond to an ectopic CM by initiating side branching. Ectopic mesenchyme is still evident as late as P21 although by P14 there is no remaining nephrogenic zone and significant cortical proximal tubular maturation (Supplementary Figure 3). While Six2 expression falls by P14, with initiation of nephron formation within ectopic CM regions, by P21, dilation of the collecting ducts of the papilla is evident, possibly due to increases in urinary filtrate production from the substantially enhanced nephron number (Supplementary Figure 3).

To discern whether transient ectopic *LIN28B* expression could affect kidney development outside of its normal expression window (after E16.5), we induced *LIN28B* at E17.5 and in newborns, time points at which expression of endogenous Lin28b is almost undetectable in the normal kidney (Fig. 1a). Six2 and Eya1 positive cells were not detected after postnatal day 2 in either the E17.5-induced or newborn-induced kidneys (Supplementary Figure 4). Collectively, these data suggest that Lin28b controls timing of kidney development, and that its expression must be precisely timed to ensure proper nephron mass. Transient over-expression of *LIN28B* can extend the period of nephrogenesis but cannot reactivate it after embryonic day 16.5.

**$LIN28B^{Wt1}$ mice exhibit enhanced renal function**. Next, we examined the effects of prolonged nephrogenesis in $LIN28B^{Wt1}$ mice (induced with dox at E16.5 for 1 day) on the function of the postnatal kidney. We first analyzed kidneys of these mice and found that in the period following *LIN28B* induction through P1 there was no significant difference between $LIN28B^{Wt1}$ and littermate control kidneys. However, with the progression of nephrogenesis, $LIN28B^{Wt1}$ kidneys gradually became dramatically and significantly larger, with a three-fold increase in weight at 2–3 months of age (Fig. 3a, b). Modest differences in the weights of other organs (heart and lungs) in 3-month-old $LIN28B^{Wt1}$ mice compared to controls did not reach statistical significance (Supplementary Figure 5a). The body weight of $LIN28B^{Wt1}$ animals was not affected for the first 2 weeks after birth. However, from P14 and until 1 month of age, $LIN28B^{Wt1}$ mice gained significantly less body weight than their littermate controls (Fig. 3c). However, once kidney development was completed in $LIN28B^{Wt1}$ mice, there was a 20% increase in body weight relative to controls at 2–3 months of age (Fig. 3c). These data indicate that *LIN28B* induction altered the developmental timing and ultimate organ size of the kidney due to prolonged nephrogenesis. To determine whether prolonged nephrogenesis in $LIN28B^{Wt1}$ mice altered nephron endowment, we calculated the nephron number by counting glomeruli in $LIN28B^{Wt1}$ kidneys compared to littermate controls. We found that $LIN28B^{Wt1}$ kidneys had a nearly two-fold increased number of glomeruli compared to 2-month-old controls, indicating increased nephron endowment in $LIN28B^{Wt1}$ animals (Fig. 3d, e). To examine whether the increased nephron number in $LIN28B^{Wt1}$ mice impacts kidney function, we measured glomerular filtration rate (GFR) and serum creatinine at 2 months of age. We found that the GFR was significantly higher in $LIN28B^{Wt1}$ mice, while serum creatinine levels were significantly lower compared to littermate controls, with no sex differences for either test (Fig. 3f, g; Supplementary Figure 5b, c). In addition, a renal function panel revealed that $LIN28B^{Wt1}$ kidneys exhibited normal levels of circulating electrolytes and proteins (Supplementary Figure 5d). These data indicate that $LIN28B^{Wt1}$ kidneys possess enhanced filtration and normal electrolyte handling. However, over a year old $LIN28B^{Wt1}$ mice showed substantive hydronephrosis with consequential loss of renal parenchyma and evidence of tubular casts (Supplementary Figure 5e). This suggested either a ureteropelvic obstruction arising from the formation of ectopic nephrons late in development or that the large volumes of urinary filtrate being produced by such an aberrantly large kidney ultimately caused a hydronephrotic pathology. Given the activity of the *Wt1-Cre* promoter in the forming nephrons, and particularly in the podocytes of the glomeruli, it is important to note an absence of apparent pathology in the glomeruli or tubular patterning and segmentation.

**Lin28b is required for the normal development of the kidney**. To test whether *Lin28b* is required for kidney development, we next generated $Lin28b^{fl/fl}$; *Wt1-Cre* KO animals (hereafter referred to as *Lin28b* KO), in which cells expressing *Wt1-Cre* lose both endogenous *Lin28b* alleles (floxed animals previously described[35]; Fig. 4a). Using optical projection tomography (OPT) for the CM-restricted transcription factor Six2, we visualized and quantified distinct CM fields or "niches" as well as the number of cells in each individual niche, in E18.5 whole kidney (Fig. 4b)[36,37]. A niche is defined as a spatially distinct cluster of CM cells and their adjacent epithelial ureteric tip. As such, niche counts also reflect the number of branchpoints of the ureteric tree[36,37]. While the number of cells in each individual niche was unaltered in kidneys of *Lin28b* KO animals relative to littermate controls, the niche count in these mice was significantly reduced, resulting in an overall decreased number of progenitor cells in the kidney (Fig. 4c, d). This reflects a decline in CM-driven ureteric branching. It has been previously established that depletion of the progenitor cell population within the MM results in premature cessation of nephrogenesis, small kidneys, low-nephron endowment, and reduced renal function[38,39]. Consistent with these studies, GFR tests revealed that *Lin28b* KO mice had significantly impaired kidney function relative to littermate controls (Fig. 4e), indicating that *Lin28b* activity is required for normal development of the mammalian kidney.

**Lin28b regulates nephrogenesis in a *let-7* dependent manner**. We have shown that transient *LIN28B* overexpression during kidney development leads to suppression of mature *let-7* species (Fig. 2c, Supplementary Figure 2a). Furthermore, we have previously shown that enforced expression of a Lin28-resistant *let-7* (*i7s*) counteracted the effect of *LIN28B* overexpression by preventing the expansion of the CM (nephron progenitors) in *LIN28B-i7s* kidneys[30]. To understand this further, we tested whether *let-7* miRNAs are functionally relevant using a genetic loss-of-function (LOF) approach. The mammalian genome encodes twelve *let-7* family members expressed from eight distinct genetic loci[40], which makes LOF studies of the *let-7* miRNA family challenging. Nevertheless, we analyzed a mouse strain harboring viable constitutive combinatorial KO of select *let-7* family members: *let-7a1*; *let-7d*; *let-7f1*[41] (hereafter referred to as *let-7* KO) (Fig. 5a; Supplementary Figure 6a). We found that while the expression of endogenous *Lin28a* and *Lin28b* mRNA did not change (Supplementary Figure 6b), *let-7* KO kidneys exhibit increased transcript levels of *Six2* and *Eya1*, similar to what was observed in $LIN28B^{Wt1}$ animals, indicating a persistence of the CM population relative to littermate controls (Fig. 5b–d). Furthermore, immunohistochemistry staining against Six2 and Lef1 revealed that progenitor cells were sustained in *let-7* KO kidneys until postnatal days P3 and P5, respectively, indicating persistence of this progenitor population one to two days longer than in littermate controls (Fig. 5c, e). While this persistent expression of CM genes is in line with prolonged nephrogenesis, in contrast to the $LIN28B^{Wt1}$ animals, the location of this persistent zone of nephrogenesis is restricted to the periphery of the kidney. Hence, the effect here is subtler and did not result in long term pathology for animals older than 1 year of age (Supplementary Figure 6c). Although the number of progenitor cells per niche was unaltered in the *let-7* KO kidneys relative to littermate controls, there was a significant increase in the niche count as shown by Six2 OPT (Fig. 5f–h). This indicates more branching and an overall increase in the number of progenitors in the whole kidney. Interestingly, unlike the $LIN28B^{Wt1}$ animals, no significant difference was observed in body weight between *let-7* KO and littermate controls (Supplementary Figure 7a) despite a

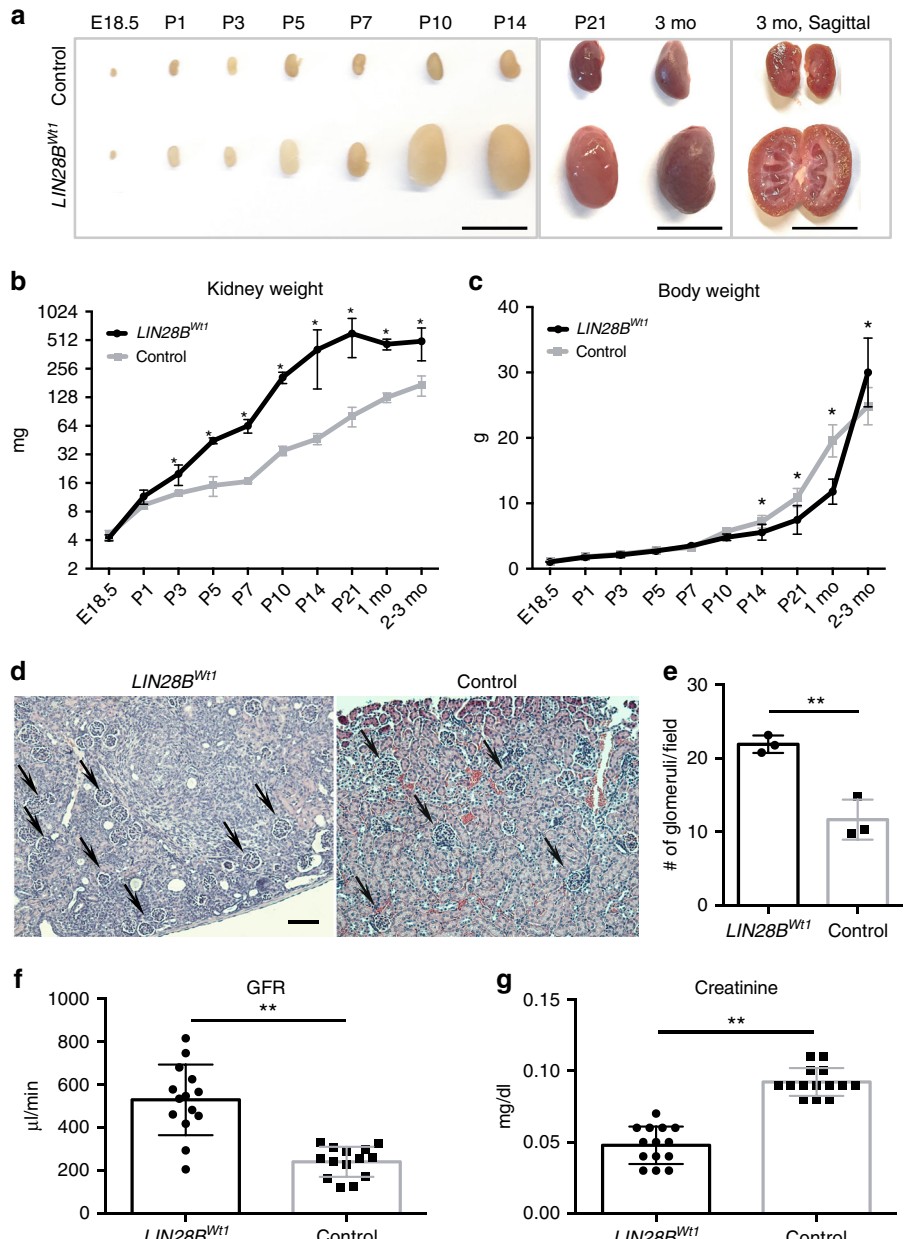

**Fig. 3** Overexpression of *LIN28B* enhances renal function. **a** Representative images of *LIN28B*^Wt1 and littermate control kidneys from E18.5 to 3 months old (m.o.) animals. Scale bar, 10 mm. **b**, **c** Kidney weight and body weight of *LIN28B*^Wt1 and littermate control mice. $n = 3–8$ for each genotype. **d** Representative hematoxylin and eosin stain (H&E) image of 1m.o. *LIN28B*^Wt1 and littermate control kidneys. Arrows point to glomerulus-like structures. Scale bar, 200μm. **e** Average number of glomeruli in 12 random fields from the kidney cortex under 10× magnification. The slides were coded and counted blindly. $n = 3$ for each group. **f**, **g** Glomerular filtration rate (GFR) and creatinine levels of *LIN28B*^Wt1 and littermate control animals (measured blindly). $n = 14$ for each genotype; each group contains mice from three different litters. Error bars represent mean ± SD. (*) $= p < 0.05$, (**) $= p < 0.01$, unpaired, two-tailed Student's *t* test

significant (more than 50%) increase in kidney weight in *let-7* KO animals from birth to adulthood (Fig. 5i) suggesting at least partial specificity of the Lin28/*let-7* axis to kidney development. Finally, *let-7* KO mice demonstrated a significant increase in GFR and nephron endowment (Fig. 5j, Supplementary Figure 7b), reduction in serum creatinine, and exhibited normal renal panel tests relative to littermate controls, similarly to *LIN28B*^Wt1 animals (Fig. 5k; Supplementary Figure 7a). Collectively, these data demonstrate that *let-7* KO results in prolonged nephrogenesis and enhanced kidney function, partially phenocopying *LIN28B* overexpression, but without aberrant pathology noted in *LIN28B* mice due to the moderate (a day or 2) delay in the cessation of CM. This

suggests Lin28b regulates nephrogenesis via suppression of *let-7* miRNA biogenesis and that proper timing of cessation of CM is essential for normal kidney function.

**Upregulation of the *Igf2/H19* locus**. To identify potential *let-7* targets we analyzed RNA sequencing (RNA-seq) data from P3 kidneys of *LIN28B*^Wt1 and *let-7* KO animals. We discovered 42 genes with overlapping expression between the two mouse models (Fig. 5l, Supplementary Figure 8), among which were two previously validated *let-7* targets, *Hmga2* and *Igf2* (Fig. 5l, Supplementary Figure 8). *Igf2* upregulation along with the

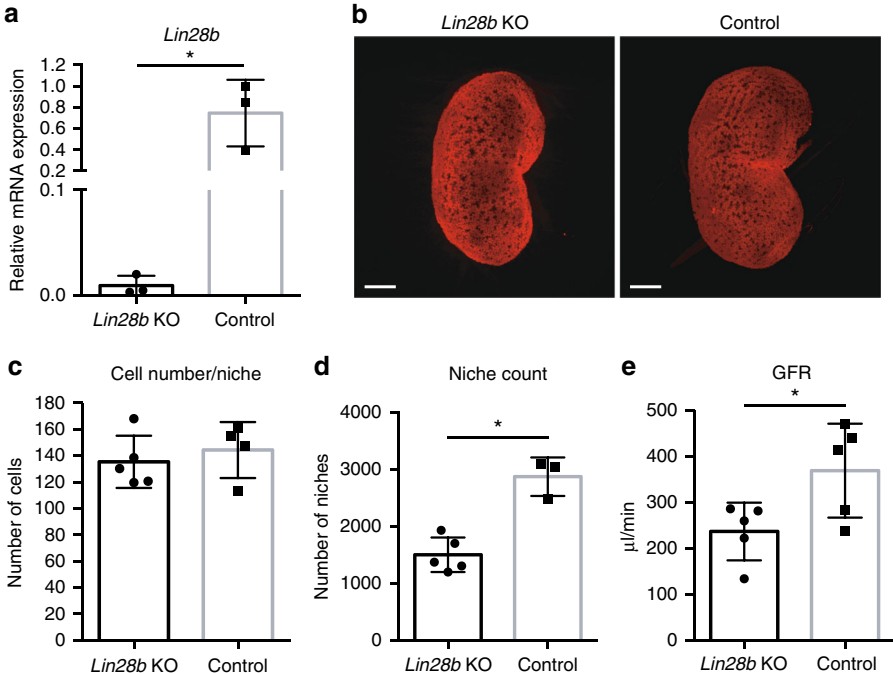

**Fig. 4** Loss of *Lin28b* leads to impaired kidney development and function. **a** Relative qRT–PCR analysis measuring levels of *Lin28b* mRNA in *Lin28b* KO and control kidneys at E18.5. $n = 3$ for each group. **b** Representative Six2 optical projection tomography (OPT) of E18.5 *Lin28b* KO and control kidneys. Scale bar, 500 μm. **c, d** The number of progenitors per niche and the number of niches, respectively, at E18.5. $n = 3-5$ for each group; each group contains mice from at least 2 different litters. **e** GFR and creatinine levels of *Lin28b* KO and littermate controls. Measured blindly in 2–3 m.o. animals. $n = 5$ for each sample type; each group contains mice from two different litters. Error bars represent mean ± SD. (*) = $p < 0.05$, unpaired, two-tailed Student's *t* test

neighboring gene *H19* is particularly interesting as *IGF2* is recognized be one of the most important Wilms tumor oncogenes. In addition, *IGF2* is commonly upregulated in Beckwith–Wiedemann and Perlman overgrowth syndromes that have high susceptibility to Wilms tumor[42,43]. Furthermore, loss of heterozygosity or imprinting on chromosome 11p15, which harbors a cluster of imprinted genes, is documented in approximately 70% of Wilms tumors, resulting in biallelic expression of *IGF2* and its neighboring *H19* gene[44,45]. Accordingly, *H19* was also significantly overexpressed in both *LIN28B^Wt1* and *let-7* KO mouse models at P3 (Fig. 5l). We further validated our RNA-seq data by carrying out qRT-PCRs for *Igf2* and *H19* spliced and unspliced mRNA transcripts in both *LIN28B^Wt1* and *let-7* KO mouse models (Supplementary Figure 9). As in our RNA-seq data, we observed a significant upregulation of *Igf2* and *H19* in spliced and unspliced mRNAs in *LIN28B^Wt1* and *let-7* KO kidneys compared to their littermate controls around the timing of cessation of nephrogenesis suggesting *Igf2* and *H19* are regulated at the transcriptional level. Taken together, these findings indicate that *Lin28/let-7* axis regulates the timing of cessation of nephrogenesis at least in part via *Igf2/H19* up-regulation.

## Discussion

The ability to form anywhere from 10,000 to 15,000 nephrons per kidney in a mouse, or 200,000 to 1.8 million nephrons per kidney in the human relies upon the self-renewal and survival capacity of renal progenitor cells during prenatal development[46]. By postnatal day 2 in mice[8] and the 36th week of gestation in humans[9], nephrogenesis terminates with the exhaustion of all remaining CM. Although existing nephrons may be repaired in response to renal injury[47], new nephrons cannot be formed during adulthood.

In mice and humans alike, IUGR correlates strongly with low-nephron endowment, which in turn, predisposes to hypertension, renal and cardiovascular diseases in the adult[11–13]. Thus, development of regenerative approaches is of particular importance to the treatment and prevention of renal disease. Several studies have attempted to recreate CM populations for in vitro nephron formation using either direct transcriptional reprogramming of somatic cells[48,49] or directed differentiation of pluripotent stem cells toward a renal progenitor fate[50–52]. However, the in vitro generation of CM is not likely to help in cases of low nephron endowment, as the anatomically complicated architecture of the kidney possesses considerable challenges to the functional integration of a stem cell-derived nephron.

Here, we report prolonged nephrogenesis, increased nephron endowment and improved kidney function due to transiently prolonged expression in utero of *Lin28b*, a heterochronic gene that has been linked to pluripotency, stem cell self-renewal, tissue metabolism, and enhanced wound healing. We show that transient overexpression of *LIN28B* in a kidney-specific and temporally defined manner delays cessation of nephrogenesis. While this increased nephron endowment yields a substantially larger organ, there is a late onset of hydronephrosis evident in these mice. This may simply result from excessive filtrate formation or represent a pathology arising from a blastemal persistence affecting the ureteropelvic junction. This pathology was not evident in *let-7* KO mice, possibly because the loss of *let-7* genes is only affecting the population of cells normally expressing *let-7* and does not result in aberrant overexpression of *LIN28* in the wider *Wt1*-expressing populations that include both the CM and the cortical stroma. The discrepancy between *let-7* KO and *LIN28B^Wt1* mouse models could also arise from the fact that Lin28b regulates other gene expression events by binding to mRNAs and modulating their translation independently of

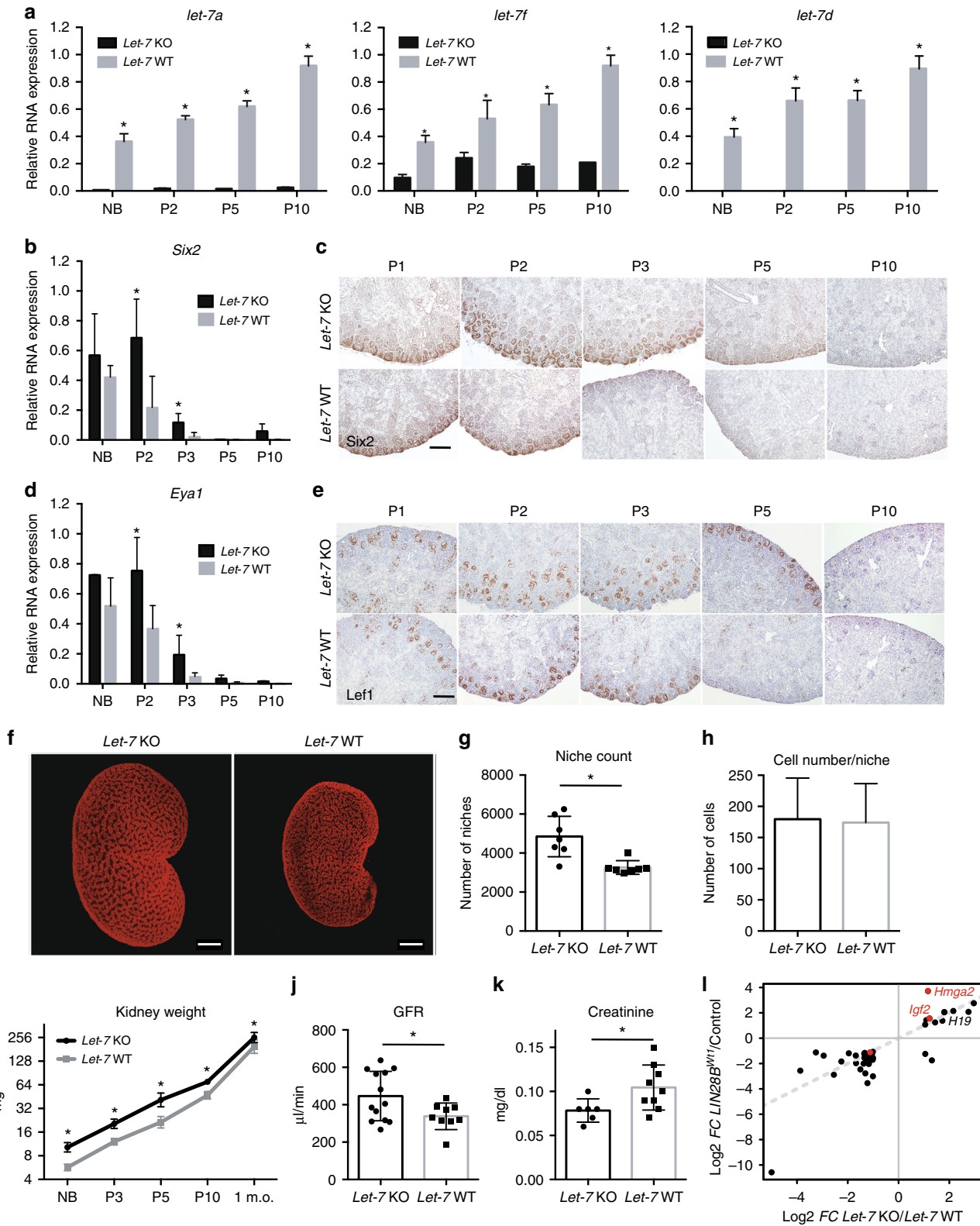

let-7[21–25], as well as from the fact that only 3 out of 12 let-7 family members were suppressed during kidney development in the let-7 KO mouse model as opposed to all of them in the LIN28B^{Wt1} animals. The observation of formation of a completely ectopic CM population within the parenchyma of the early postnatal kidney in the LIN28B^{WT1} overexpression model is intriguing.

While this appeared to resolve into nephrons with time, the pathology presented as for nephroblastomatosis/persistent nephrogenic rests, a condition in which there is postnatal persistence of a "blastema" without obligate oncogenic transformation[32,53]. This suggests a capacity for LIN28B to convert renal mesenchyme to a CM phenotype. This may have

**Fig. 5** *Let-7* KO prolongs nephrogenesis and enhances kidney function. **a, b, d** Relative qRT–PCR analysis measuring the levels of mature *let-7a,-f,-d*; Six2; Eya1 RNAs, respectively, in KO mice and wild type (WT) littermates at the indicated developmental time points. $n = 2$–6 for each genotype.
**c, e** Representative immunohistochemistry staining against Six2 and Lef1, respectively, in *let-7* KO mice and WT littermates. **f** Representative Six2 OPT of P1 *let-7* KO and WT kidneys. Bar, 500 μm. **g, h** The number of niches and the number of progenitors per niche, respectively, at P1. $n = 7$ for each genotype; each group contains mice from three different litters. **i** Kidney weight of *let-7* KO mice and WT littermates. Log2 scale. $n = 2$–6 for each genotype. **j, k** GFR and creatinine levels of the *let-7* KO and WT littermate controls. Measured blindly in 2–3 m.o. animals. $n = 6$–13 for each genotype; each group contains mice from 2 to 3 different litters. **l** Scatter plot of differentially expressed genes between same-aged (P3) LIN28B$^{Wt1}$ and *Let-7* KO mouse models measured by RNA-seq. Differentially expressed genes were identified from two mice models individually with $p$ value $< 0.01$. Genes with $p$ value $< 0.01$ in both conditions were plotted on Log2 scale. Predicted *let-7* targets from target scan server are indicated in red. $n = 3$ biological replicates from two different litters for each model. Error bars represent mean ± SD. (*) $= p < 0.05$, unpaired, two-tailed Student's *t* test

---

implications in studies looking to recreate or prolong a nephro-genic population in vitro.

Both overexpression of *LIN28B* and KO of *let-7* results in an upregulation of *Igf2* and *H19* genes at the transcriptional level. While the locus of *H19/Igf2* has been shown to be co-regulated by *cis*-regulatory elements in a tissue-specific manner, recent studies have also indicated that the *H19* antisense RNA transcript can also *trans*-activate the *Igf2* promoter[54]. We have demonstrated in our study the up-regulation of both spliced and unspliced *Igf2* and *H19* transcripts. This observation begs the question of whether this is a direct consequence of the ability of Lin28b or *let-7* to bind to the *H19* RNA antisense transcript knowing that both Lin28 and *let-7* are capable of binding to *Igf2* mRNA[21].

In this study, we show definitively that kidney specific loss of *Lin28b* results in significantly impaired kidney development, confirming the crucial role of this gene in proper developmental timing of nephrogenesis. We have linked the activity of Lin28b in kidney development to its ability to modulate the production of the *let-7* family of miRNAs and shown that Lin28b/*let-7* axis regulates the cessation of nephrogenesis possibly via upregulation of the growth-promoting gene *Igf2*. Modulating cessation of nephrogenesis by delaying it by a day or two (similar to our *let-7* KO phenotype) would be an ultimate goal in such studies as it would enhance nephrogenesis and increase the nephron mass without pathological changes (Fig. 6). Indeed, restricting that overactivity to the remaining CM within the peripheral nephro-genic zone, rather than allowing aberrant formation of ectopic nephrons within a wider mesenchymal population, will also be critical to avoid concomitant pathology with such a strategy.

## Methods
**Animals**. All animal work was done in accordance with IACUC guidelines at the ARCH facility in Children's Hospital Boston. The generation and maintenance of *Col1a-TRE-LIN28B* and *Lin28b*$^{fl/fl}$ animals was previously described briefly, flag-tagged human *Lin28* open reading frame was cloned into pBS plasmid and targeting was performed into V6.5 ES cells containing *M2-rtTA* targeted to the *Rosa26* locus. For *lin28b* conditional KO mice, PCR fragments of both gene loci were cloned into a plasmid having two loxP cassettes and a PGK-Neo cassette flanked with frt sequences, and targeting was performed into V6.5 embryonic stem cells (ESCs). The *let-7* KO strain was a gift from Antony Rodriguez. The *Wt1-Cre* mice were contributed by the laboratory of Dr. William Pu at the Boston Children's hospital. For transgene induction, 1 g/L doxycycline (Sigma) was administered to the drinking water at different time points to induce *LIN28B*. Weaning mice were genotyped via ear clippings processed by Transnetyx. Both males and females were examined throughout this study and no observable differences were seen.

**Quantitative RT-PCR**. RNA was isolated by TRIzol from whole kidneys and reverse-transcribed using a miScriptII RT kit (Qiagen, #218161). Relative mRNA expression was measured by qPCR using the ΔΔCT method with the following primers: mSix2 (forward primer, 5′-GCAAGTCAGCAACTGGTTCA-3′;reverse primer, 5′-CTTCTCATCCTCGGAACTGC-3′), mEya1 (forward primer, 5′-TTTCCCTGGGACTACGAATG-3′; reverse primer, 5′-GGAAAGCCATCTG TTCCAAA-3′), mbActin (forward primer, 5′-TACTCCTGCTTGCTGATCCAC-3′; reverse primer, 5′-CAGAAGGAGATTACTGCTCTGGCT-3′); and hLIN28B (for-ward primer, 5′-GCCCCTTGGATATTCCAGTC-3′; reverse primer, 5′-TGACT CAAGGCCTTTGGAAG-3′); mLin28b (forward primer, 5′-TTTGGCTGAGGA GGTAGACTGCAT-3′; reverse primer 5′-ATGGATCAGATGTGGACTGTGCGA-

3′); mLin28a (forward primer, 5′-AGCTTGCATTCCTTGGCATGATGG-3′; reverse primer- 5′-AGGCGGTGGAGTTCACCTTTAAGA-3′). Absolute quantifi-cation PCR was performed by using DNA standards ordered from IDT for amplicons of mLin28a and Lin28b primers. For qRT-PCR of mature and precursor *let-7* miRNAs, we used Qiagen miScript target as described by the manufacturer.

**Immunoblot analysis**. Whole kidneys (from E12.5 to adulthood) were dissected and then lysed in RIPA buffer (Pierce) supplemented with protease inhibitor cocktail (Roche) and phosphatase inhibitor cocktail (Roche). Lysates were loaded and run on the 12% polyacrylamide gel (Bio-Rad) in 5× Laemmli sample buffer and transferred to a nitrocellulose membrane (GE Healthcare). The membrane was blocked for 1 h in PBST containing 5% milk and subsequently probed with primary antibodies overnight at 4 °C. After 1-h incubation with sheep anti-mouse or donkey anti-rabbit HRP-conjugated secondary antibody (GE Healthcare), the protein level was detected with standard ECL reagents (Thermo Scientific). Antibodies used: anti-α/β-tubulin (1:1000, Cell Signaling, #2148), anti-Lin28a (1:1000, Cell Signal-ing, #3978), anti-Lin28b (1:1000, mouse preferred) (Cell Signaling, #5422), anti-Six2 (1:1000, Proteintech Group, #11562-1-AP). Uncropped scans included in Supplementary Figure 10.

**Histological analysis**. Whole kidneys were fixed in 10% formalin overnight at room temperature, then placed in 70% ethanol and embedded in paraffin. Slides were dewaxed with xylene and rehydrated through a series of washes with decreasing percentages of ethanol. Antigen retrieval was performed in 10 mM sodium citrate buffer (pH 6.0) by placement in decloaking chamber for 45 min at 95 °C. Slides were treated with 10% hydrogen peroxide to inhibit endogenous peroxidase activity. After blocking with 5% goat or rabbit serum (VECTASTAIN ABC kit #PK-6101), slides were incubated with primary antibody overnight at 4 °C and secondary antibody for 30 min at room temperature. Detection was performed with the VECTASTAIN Elite ABC Kit and DAB Substrate (Vector Laboratories, SK-4100). Sections were counterstained with hematoxylin for 20–30 s then dehy-drated in increasing concentrations of ethanol before a 5-min incubation in xylene followed by mounting. Antibodies used: anti-LIN28B (1:250, Cell Signaling, #4196), anti-Six2 (1:250, Proteintech Group, #11562-1-AP), and anti-Lef1 (1:250, Cell Signaling, #2230).

**Immunofluorescence and image analysis**. Whole kidneys were fixed in 4% PFA for half an hour at 4 °C then placed in PBS. Whole mount immunofluorescence, confocal microscopy, and OPT was carried out according to published protocols, briefly, after initial in vivo cell labeling with the nucleoside analog 5-ethynyl-2'-deoxyuridine (EdU) and tissue-specific antibodies, OPT and confocal microscopy are used to image the developing kidney. These imaging data then inform a second analysis phase that quantifies (using Imaris and Tree Surveyor software), models and integrates these events at a cell and tissue level in 3D space and across developmental time. Cell counts per niche (confocal) and niche counts (OPT) were performed as reported[36]. Antibodies used: rabbit anti-Six2 (1:600, Proteintech Group, #11562-1-AP), anti-rabbit Alexa Fluor-568 conjugated secondary antibody (1:300, Life Technologies).

**Blood analysis**. Renal panel tests performed on an Abaxis VetScan VS2 chemistry analyzer. Serum creatinine measured using isotope dilution LC-MS/MS in the O'Brien Core Center for Acute Kidney Injury Research, the University of Alabama at Birmingham School of Medicine.

**Glomeruli number count**. To compare the nephron number between LIN28B$^{Wt1}$ and control mice we count the number of glomerulus-like structures in 12 random fields from the kidney cortex in H&E sections under 10× magnification.

**Measurement glomerular filtration rate**. GFR was measured using a high-throughput method described previously[50]. Fluorescein isothiocyanate (FITC)-sinistrin (Fresenius Kabi, Linz, Austria) was administered to conscious mice under light anesthesia, isoflurane, via tail vein injections. Blood was collected from a small

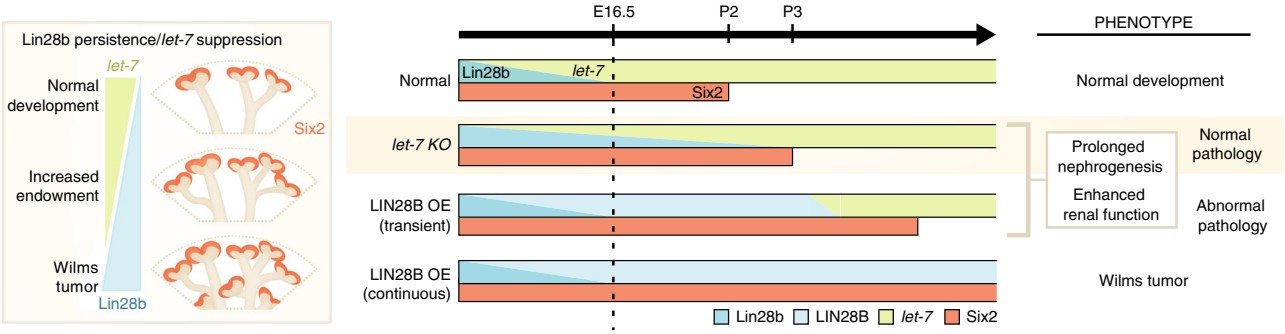

**Fig. 6** Schematic of enhanced nephrogenesis by Lin28/*let-7* pathway. During normal kidney development Six2+ cap mesenchyme (CM) are sustained in the outer nephrogenic zone of the kidney until postnatal day 2 in mice, after which time all remaining CM cells undergo a synchronous wave of differentiation to establish the final number of nephrons that will persist lifelong in the adult. Lin28b/*let-7* axis controls the timing and duration of kidney development such that overexpression of LIN28B or suppression of *let-7* miRNAs prolongs the period of nephrogenesis. Nephrogenesis prolonged just for a few days like in *let-7* KO results in enhanced renal function and normal physiology, while longer persistence of nephrogenesis akin to *LIN28B* transient and continuous overexpression leads to abnormal pathology and complications like hydronephrosis and Wilms tumor in later life

tail snip at 3, 7, 10, 15, 35, 55, and 75 min postinjection for the determination of FITC concentration by fluorescence. GFR was calculated by a two-phase exponential decay model[50].

**RNA-seq library preparation and data analysis**. RNA from each kidney was isolated using Trizol and treated with DNase. Sequencing libraries were generated using a SMARTer Ultra Low RNA kit (V3) and sequenced on the HiSeq 2500 machine (Illumina). Then, reads were analyzed using our gene expression pipeline[41]. Briefly, reads were aligned to mouse transcriptome and differentially expressed genes were identified using an edgeR package.

**Statistical analysis**. Data is expressed as mean ± SD. Unpaired *t* test with two-tailed distribution and Welch's correction was calculated using Prism (GraphPad Prism) to determine *p* values. Statistical significance is displayed as $p < 0.05$ (*), $p < 0.01$ (**).

**Gene nomenclature**. Human gene—capital italic (*LIN28B*), human protein—capital (LIN28B), mouse gene—first letter capital, italic (*Lin28b*). Mouse protein—first letter capital (Lin28b).

## Data availability
The raw data for the RNAseq have been deposited in the NCBI Gene Expression Omnibus (GEO) database under accession code GSE117510. All other relevant data that support the findings of this study are available from the corresponding authors upon reasonable request.

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

## Acknowledgements

We are grateful to Antony Rodriguez for the gift of the *let-7* knockout animals. We thank ARCH for their assistance with maintenance of the animal colony. We would also like to thank Rod Bronson from the Rodent Histopathology core at Harvard Medical School for initial mouse tissue pathology and Tom Forbes, Royal Children's Hospital, Melbourne, and Sarah Walton, Monash University, Melbourne, for advice around pathology. A.N.C. was supported by a DECRA fellowship from the Australian Research Council and M.H.L. is a Senior Principal Research Fellow of the National Health and Medical Research Council. This project was supported by NIH F99 CA212487 predoctoral fellowship to A.V.Y., Burroughs Wellcome post-doctoral enrichment scholarship to J.K.O., NHMRC Project grant (APP1063989) to M.H.L., and NIH RO1-GM107536 and administrative supplement 03S1 to G.Q.D.

## Author contributions

A.V.Y. and G.Q.D. conceived the project. A.V.Y. and J.K.O. interpreted data, designed experiments, and performed them with assistance from P.S., M.A.K., D.S.P., H.M., D.A.R. M.J.C. helped with dissection of mouse embryonic kidneys. A.H. performed RNA-seq library preparation and data analysis. A.N.C. and S.B.W. conducted 3D immunofluorescence and quantitative image analysis of whole mount kidneys. M.H.L. interpreted data and analyzed kidney pathology. A.V.Y., J.K.O., G.Q.D., and M.H.L. wrote the manuscript with input from all authors.

## Additional information

**Competing interests:** G.Q.D. is a founder, owns equity, and receives consulting fees from 28/7 therapeutics, a biotechnology company seeking to develop drugs targeting the LIN28/let-7 pathway for the treatment of cancer. The remaining authors declare no competing interests.

