## [Peer Review File · Nature Communications]

Reviewers' comments:

Reviewer #1 (Remarks to the Author):

This is potentially a very interesting study showing, as hypothesised, that the Lin28 let-7 axis regulates the timing of cessation of nephrogenesis. In the main, the data are of high quality and support the conclusions.

The most dramatic finding is that a pulse of Lin28 overexpression at E16.5 in the metanephric mesenchyme leads to the formation of ectopic fields of nephron progenitors. This is associated with expansion of the cap mesenchyme deep into the parenchyma. Interestingly, this appears to initiate new lateral bud formation. Remarkably, the resulting adult mice have double the number of nephrons and enhanced kidney function-though by a year the kidneys develop hydronephrosis. Strikingly, only a minimal effect is observed when Lin28 is induced a day later.

Major Points

1. The main assumption throughout the paper, and partially supported by the data is that these effects are exerted through downregulation of the Let-7 family of miRNAs. A priori this is a reasonable assumption given the fact that one of the main functions of Lin28 is to inhibit the production of Let-7s which in turn downregulate Lin28 in a negative feedback loop.

The authors provide several lines of evidence to support this hypothesis

Firstly, as the levels of Lin28b decrease during nephrogenesis the processing of particular Let-7 members increases. Secondly, the induced overexpression of Lin28 in the kidney mesenchyme leads to a reduction in the levels of all Let7 miRNAs tested. Thirdly, mice with KO of 3 Let-7 members have a phenotype that partially overlaps that for the Lin28 gain of function. Importantly, and potentially of clinical relevance, the mice have an increased nephron endowment with an increase in Glomerular Filtration rate but no obvious pathology. It is not surprising the phenotypes did not completely overlap as only 3 Let-7 genes were deleted out and these were constitutive knockouts (the authors might have discussed this).

In spite of this strong circumstantial evidence the authors cannot formally claim as in the title on page 10 " Lin28b regulates nephrogenesis in a let-7 dependent manner". To prove this they should determine whether adding back a Lin28-resistant Let-7 reverses the phenotype. Professor Daley's group carried out such an experiment successfully in their *Genes and Dev* paper in 2014. I don't know if there is a reason why this same approach could not be used here. In an ideal world such an experiment should be carried out.

2. To investigate downstream mechanism the authors carry out RNA seq on the kidneys of the Lin28 overexpressing and Let-7 KO mice to look for upregulated RNAs known to be direct let-7 targets in both situations. A number of such RNAs are identified but the analysis is very superficial and unsatisfactory. One of the upregulated RNAs is that for Igf2. Given a burgeoning number of studies showing the importance of increased Igf2 levels in Wilms' Tumour, the authors focus their attention on this mRNA. There are several issues here. Firstly, the mRNA for another key proliferation-promoting protein, Hmga2 appears to be much more significantly induced. In 2017, a study in *Oncotarget* supported a Lin28/let-7/Hmga2 axis in Wilms' tumor. Another important consideration is that Lin28 has been shown to bind directly to Igf2 mRNA, regulating its translation, but not mRNA level. To help clarify the situation more rigorously, I suggest the authors carry out quantitative RTPCR for Igf2, Hmga2 and any other plausible targets, on kidneys from the different mouse models, not just at one postnatal stage, but also at late and relevant stages of embryogenesis. If Igf2 holds up as a candidate (and/or Hmga2) the authors should measure the level of unspliced transcripts to rule out transcriptional effects - and measure protein levels assuming the antibodies are good.

Minor points

1. On page 6 the authors suggest, based on developmental profiles, that specific let-7 subsets may be responding to Lin28 during development. Is there any biochemical or structural information that might provide an explanation for the specificity. On the other hand, when Lin28 is overexpressed all the let7 RNAs are downregulated, perhaps due to massive overexpression. Perhaps the authors could discuss.
2. Lin28 levels should be measured in the kidneys from the Let-7 KO pre and postnatally.

--

Reviewer #2 (Remarks to the Author):

Daley and colleagues present a study on the role of Lin28/let-7 axis in kidney development. They observe some level of negative correlation between Lin28b and let-7 microRNAs during formation of nephrons. Then, they report that transient over-expression of Lin28b prolongs nephrogenesis and enhances kidney function in young animals, albeit causing pathology in older mice. They also use Lin28 kidney-specific KO to show that Lin28b is required for normal kidney development. Finally, they demonstrate that knock out of several members of the let-7 family partially phenocopies Lin28b over-expression.

In general, the study is interesting as it highlights the importance of spatiotemporal expression of RNA-binding protein (Lin28b) and regulatory RNAs (let-7 family) in mammalian kidney development. However, my enthusiasm is somehow dampened by the fact that the authors seem to focus entirely on the Lin28/let-7 axis, forgetting that Lin28 regulates many other gene expression events and microRNAs. This is most visible in partial overlap between Lin28a over-expression and let-7 KO phenotypes.

My specific questions and suggestions are as follow:

1. Pre-let-7c-2 escapes Lin28-mediated regulation (Triboulet R Cell Reports 2015), and yet let-7c follows the same pattern as all other let-7 in kidney development. Why is that?
2. What is the correlation coefficient between Lin28b and let-7 microRNAs levels? The biggest drop of Lin28b is between E16.5 and E17.5 but let-7 levels don't change in this period. To me, it looks like there are other regulators (or cofactors) of let-7a biogenesis that could play a role in kidney development. This should be highlighted and discussed accordingly.
3. 'pre-let-7-a2, pre-let-7-b1, pre-let-7-c1, pre-let-7-d, pre-let-7-e, have the same pattern of expression as their mature family members'. Is there a possibility of common transcription and post-transcriptional regulation of mature let-7 levels as seen for let-7 in Van Wynsberghe PM Nat Struct Mol Biol 2011 and miR-9 in Nowak JS Nat Commun 2014?
4. Lin28b is known to regulate pri-let-7 to pre-let-7 processing. What is the mechanism of Lin28b regulation in developing kidney?
5. Figure 2B presents immunostaining of Lin28b in LIN28B_wt1 mouse. It would be much more informative to present a western blot to compare the endogenous levels at E12.5 or E13.5 in the wild type mouse to the levels of over-expressed Lin28b. What is the fold increase of Lin28a? Is it within the physiological range?
6. 12 out of 17 predicted let-7 targets are upregulated in Lin28B and let-7 KO mouse models. 5 out of 17 predicted let-7 targets are downregulated in Lin28B and let-7 KO mouse models. What does it mean? Could Lin28B or let-7 have positive effects on expression of these 5 genes? This could be easily tested with luciferase reporter assays.

7. The discrepancy between let-7 KO and Lin28b over-expressing mice could arise from the fact that Lin28 regulates many other microRNAs and gene expression events. This should be adequately discussed and referenced with the existing literature.

--

Reviewer #3 (Remarks to the Author):

This manuscript by Yermalovich et al is a follow up study of an earlier paper published in 2014. Compared to the previous study, the authors demonstrated that Lin28b, but not Lin28a, is inversely correlated with let-7 microRNA expression during kidney development, suggesting its predominant role in this process. Through an extensive set of experiments using transgenic mouse models, they demonstrated that transient overexpression of LIN28B at E16.5 led to prolonged kidney development and enhanced kidney function. This effect can be recapitulated in let-7 KO. In contrast, depletion of LIN28B resulted in impaired kidney function. The authors convincingly showed that changes of let-7 and downstream targets upon perturbation of LIN28B. These include Igf2/H19, which was previously associated with Wilms tumorigenesis, although details were not investigated. This is an interesting paper that demonstrated the endogenous function of LIN28B during development, which is currently not clear in the field.

Minor points:

1. p4. This study did not directly compare the importance of Lin28a and Lin28b in terms of their function in development (except for analysis of gene expression). I would suggest the authors tone down their conclusion "... implicating Lin28b (and not Lin28a) as playing the predominant role in nephrogenesis.
2. page 6, microRNA names should be double checked, e.g., pre-let-7b1, g1, i1 should be pre-let-7b, g, i.
3. page 9, Lin28fl/fl should be Lin28b fl/fl.
4. Fig. S8, LIN28Bwt1 3 appears to be outlier. The authors might identify a more robust set of targets by excluding the outlier?

Reviewers' comments:

Reviewer #1 (Remarks to the Author):

This is potentially a very interesting study showing, as hypothesised, that the Lin28 let-7 axis regulates the timing of cessation of nephrogenesis. In the main, the data are of high quality and support the conclusions. The most dramatic finding is that a pulse of Lin28 overexpression at E16.5 in the metanephric mesenchyme leads to the formation of ectopic fields of nephron progenitors. This is associated with expansion of the cap mesenchyme deep into the parenchyma. Interestingly, this appears to initiate new lateral bud formation. Remarkably, the resulting adult mice have double the number of nephrons and enhanced kidney function-though by a year the kidneys develop hydronephrosis. Strikingly, only a minimal effect is observed when Lin28 is induced a day later.

Major Points

1. The main assumption throughout the paper, and partially supported by the data is that these effects are exerted through downregulation of the Let-7 family of miRNAs. A priori this is a reasonable assumption given the fact that one of the main functions of Lin28 is to inhibit the production of Let-7s which in turn downregulate Lin28 in a negative feedback loop. The authors provide several lines of evidence to support this hypothesis. Firstly, as the levels of Lin28b decrease during nephrogenesis the processing of particular Let-7 members increases. Secondly, the induced overexpression of Lin28 in the kidney mesenchyme leads to a reduction in the levels of all Let7 miRNAs tested. Thirdly, mice with KO of 3 Let-7 members have a phenotype that partially overlaps that for the Lin28 gain of function. Importantly, and potentially of clinical relevance, the mice have an increased nephron endowment with an increase in Glomerular Filtration rate but no obvious pathology. It is not surprising the phenotypes did not completely overlap as only 3 Let-7 genes were deleted out and these were constitutive knockouts (the authors might have discussed this). In spite of this strong circumstantial evidence the authors cannot formally claim as in the title on page 10 "Lin28b regulates nephrogenesis in a let-7 dependent manner". To prove this they should determine whether adding back a Lin28-resistant Let-7 reverses the phenotype. Professor Daley's group carried out such an experiment successfully in their Genes and Dev paper in 2014. I don't know if there is a reason why this same approach could not be used here. In an ideal world such an experiment should be carried out.

Thank you very much for your suggestion. We have already shown in our Genes and Development paper in 2014 that enforced expression of a Lin28-resistant *let-7 (i7s)* counteracts the effect of *LIN28B* overexpression by preventing the expansion of the cap mesenchyme (nephron progenitors) in *LIN28B-i7s* kidneys. In our current and previous studies we utilized the same model (*TRE-LIN28B; lox-STOP-lox-rtTA*) with a *Wt1-Cre* driver. Therefore, repeating this experiment would feel redundant and take many months to perform. In addition, in the current study we have shown that

deletion of 3 out of 12 family members of the *let-7* family of miRNAs recapitulates *LIN28B* phenotype firmly linking *Lin28* and *let-7* together as regulators of the timing of cessation of nephrogenesis. We have revised the text to emphasize these points and hope that this is sufficient in the eyes of the reviewer.

2. To investigate downstream mechanism the authors carry out RNA seq on the kidneys of the *Lin28* overexpressing and *Let-7* KO mice to look for upregulated RNAs known to be direct *let-7* targets in both situations. A number of such RNAs are identified but the analysis is very superficial and unsatisfactory. One of the upregulated RNAs is that for *Igf2*. Given a burgeoning number of studies showing the importance of increased *Igf2* levels in Wilms' Tumour, the authors focus their attention on this mRNA. There are several issues here. Firstly, the mRNA for another key proliferation-promoting protein, *Hmga2* appears to be much more significantly induced. In 2017, a study in *Oncotarget* supported a *Lin28/let-7/Hmga2* axis in Wilms' tumor. Another important consideration is that *Lin28* has been shown to bind directly to *Igf2* mRNA, regulating its translation, but not mRNA level. To help clarify the situation more rigorously, I suggest the authors carry out quantitative RTPCR for *Igf2*, *Hmga2* and any other plausible targets, on kidneys from the different mouse models, not just at one postnatal stage, but also at late and relevant stages of embryogenesis. If *Igf2* holds up as a candidate (and/or *Hmga2*) the authors should measure the level of unspliced transcripts to rule out transcriptional effects - and measure protein levels assuming the antibodies are good.

We thank the reviewer for this valuable suggestion. We re-analyzed our RNA seq as we found an error in our prior differential gene analysis, in which one of the sample labels was swapped during the blinded analysis by our bioinformatician. We corrected this error and re-ran the analysis, identifying a more robust set of overlapping genes (Fig. 5L, Supplemental Fig.S8). We discovered 42 overlapping genes, only 3 of which were predicted *let-7* targets (Fig. 5L, Supplemental Fig.S8). Two out of three, *Hmga2* and *Igf2*, are upregulated in *LIN28B^{Wt1}* and *let-7* KO mouse models, while one, *Nuak2*, is down regulated. Both *Hmga2* and *Igf2* are well known and characterized *let-7* targets that have been implicated in Wilms tumor (and therefore in kidney development). *Nuak2* is a predicted *let-7* target according to Target Scan, but not validated and its role in kidney development has not been documented. Consequently, in this study we focus on *Igf2* given, as you mentioned, the burgeoning number of studies showing the importance of increased *Igf2* levels in Wilms Tumor and hence kidney development.

As you suggested, we further validated our RNA seq data by carrying out quantitative RT-PCRs for *Igf2* and *H19* spliced and unspliced mRNA transcripts in both *LIN28B^{Wt1}* and *let-7* KO mouse models (Supplemental Figure S9). As in our RNA seq data, we observed a significant upregulation of *Igf2* and *H19* in spliced and unspliced mRNAs in *LIN28B^{Wt1}* and *let-7* KO kidneys compared to their littermate controls around the timing of cessation of nephrogenesis suggesting increased transcription of the *Igf2/H19* locus.

Minor points

1. On page 6 the authors suggest, based on developmental profiles, that specific let-7 subsets may be responding to Lin28 during development. Is there any biochemical or structural information that might provide an explanation for the specificity.

Unfortunately, we do not have any biochemical or structural information that might explain this specificity. This is a fascinating conundrum in the field and an important future extension of research.

On the other hand, when Lin28 is overexpressed all the let7 RNAs are downregulated, perhaps due to massive overexpression. Perhaps the authors could discuss.

Thank you very much for this suggestion. We have now addressed this point in the text with the following statement: “ ...Transgene induction led to persistence of *LIN28B* mRNA and protein levels out to postnatal day 5 (P5) in *LIN28B^{Wt1}* kidneys (Fig. 2A,B), resulting in a consequent suppression of all *let-7* miRNAs during this period due to high level overexpression of ectopic *LIN28B* (Fig 2C; Supplemental Fig.S2A).

2. Lin28 levels should be measured in the kidneys from the let-7 KO pre- and postnatally.

We have analyzed the expression of endogenous *Lin28a* and *Lin28b* mRNA in the *let-7* KO mouse model at different and relevant time points of kidney development and observe no change in transcript levels between *let-7* KO and littermate controls at all the time points tested (Supplemental FigureS6.B).

Reviewer #2 (Remarks to the Author):

Daley and colleagues present a study on the role of Lin28/let-7 axis in kidney development. They observe some level of negative correlation between Lin28b and let-7 microRNAs during formation of nephrons. Then, they report that transient over-expression of Lin28b prolongs nephrogenesis and enhances kidney function in young animals, albeit causing pathology in older mice. They also use Lin28 kidney-specific KO to show that Lin28b is required for normal kidney development. Finally, they demonstrate that knock out of several members of the let-7 family partially phenocopies Lin28b over-expression.

In general, the study is interesting as it highlights the importance of spatiotemporal expression of RNA-binding protein (Lin28b) and regulatory RNAs (let-7 family) in mammalian kidney development. However, my enthusiasm is somehow dampened by the fact that the authors seem to focus entirely on the Lin28/let-7 axis, forgetting that Lin28 regulates many other gene expression events and microRNAs. This is most visible in partial overlap between Lin28a over-expression and let-7 KO phenotypes.

My specific questions and suggestions are as follow:

1. Pre-let-7c-2 escapes Lin28-mediated regulation (Triboulet R Cell Reports 2015), and yet let-7c follows the same pattern as all other let-7 in kidney development. Why is that?

This is a very interesting question. Triboulet and colleagues indeed showed that *pre-let-7c-2* bypasses LIN28 mediated repression. However, this was only tested in Hela cells (cervical tumor cell lines) and mouse embryonic stem cells (mESCs). It is possible that Lin28 proteins regulate *let-7* microRNAs during mouse development differently compared to Hela or mESCs. Triboulet and colleagues themselves concluded in their discussion that “It will be interesting to explore the full range of physiological contexts where that bypass is relevant” acknowledging the fact that their findings are context dependent.

2. What is the correlation coefficient between Lin28b and let-7 microRNAs levels? The biggest drop of Lin28b is between E16.5 and E17.5 but let-7 levels don't change in this period. To me, it looks like there are other regulators (or cofactors) of let-7a biogenesis that could play a role in kidney development. This should be highlighted and discussed accordingly.

Thank you for this keen observation. One of the other known regulators (or cofactor) of *let-7* biogenesis during kidney development is Lin28a. In early nephrogenesis, at E12.5 and E13.5, both Lin28 paralogs are present and are possibly down-regulating *let-7* processing. At E14.5 Lin28a expression ceases and mature *let-7* microRNAs start to increase. However, the most significant change in the mature *let-7* miRNAs is observed between E14.5 and E16.5, which correlates with

the gradual decrease of Lin28b expression. As suggested, we have now addressed this point in the result section.

3. 'pre-let-7-a2, pre-let-7-b1, pre-let-7-c1, pre-let-7-d, pre-let-7-e, have the same pattern of expression as their mature family members'. Is there a possibility of common transcription and post-transcriptional regulation of mature let-7 levels as seen for let-7 in Van Wynsberghe PM Nat Struct Mol Biol 2011 and miR-9 in Nowak JS Nat Commun 2014?

This is a very interesting question. While there is certainly a possibility of common transcriptional and post-transcriptional regulation of *let-7* levels during kidney development, the analysis done by “Van Wynsberghe PM Nat Struct Mol Biol 2011” was in worm, where you only have one Lin-28 and one *let-7* and in human ESC which represent a single time point during embryonic development. With regards to the “Nowak JS Nat Commun 2014”, again this work was done during the in vitro differentiation of the P19 embryonic carcinoma cell line, not throughout development. As such it would be difficult to compare the actual biochemistry of how these molecules are regulated throughout maturation in these settings.

4. Lin28b is known to regulate pri-let-7 to pre-let-7 processing. What is the mechanism of Lin28b regulation in developing kidney?

Indeed, there is a study that suggests that Lin28a is predominantly localized in the cytoplasm, where it recruits TUT4/7 to oligo-uridylate *pre-let-7* and prevents Dicer processing, whereas Lin28b is predominantly localized in the nucleolus where it sequesters *pri-let-7* away from Drosha and DGCR8 processing (Piskounova E Cell 2011). However, all Lin28 proteins (in mammals as well as in *C. elegans*) have a putative nucleolar localization signal and are able to enter both the nucleus and cytoplasm and have been also shown to bind to both *pri-let-7* and *pre-let-7* and block their processing into mature miRNA (Viswanathan SR Science 2008, Heo I Molecular Cell 2008, Rybak A Nature Cell Biology 2008).

We don't know the exact mechanism of Lin28b regulation in the developing kidney. Lin28b can be regulated at least in part by *let-7* microRNA family, that are known to bind to the 3' UTR of Lin28 mRNA, negatively regulating its expression and forming a bistable switch that is conserved throughout evolution from worms to humans (Viswanathan SR Science 2008, Heo I Molecular Cell 2008, Newman MA RNA. 2008). The mechanism of transcriptional regulation of Lin28 has not been well defined; thus, we have discussed what is known in the intro and cited these papers.

5. Figure 2B presents immunostaining of Lin28b in LIN28B_wt1 mouse. It would be much more informative to present a western blot to compare the endogenous levels at E12.5 or E13.5 in the wild type mouse to the levels of over-expressed Lin28b. What is the fold increase of Lin28a? Is it within the physiological range?

Unfortunately, we are not able to directly compare the endogenous levels of Lin28b at E12.5 or E13.5 in the wild type mouse to the levels of over-expressed LIN28B by a western blot as our transgenic mouse model is human LIN28B driven (and not mouse). We looked at the endogenous mRNA levels of *Lin28a* and *Lin28b* in our LIN28B^{Wt1} mouse model from E18.5 to P21 and observe no change in the expression between LIN28B^{Wt1} mice and their littermate controls at all the time points tested. Most of the *Lin28a* and *Lin28b* mRNA levels were at undetected levels for each group and dramatically (almost 100 fold) lower compared to the endogenous mRNA levels at E12.5 in the wild type mouse.

6. 12 out of 17 predicted let-7 targets are upregulated in Lin28B and let-7 KO mouse models. 5 out of 17 predicted let-7 targets are downregulated in Lin28B and let-7 KO mouse models. What does it mean? Could Lin28B or let-7 have positive effects on expression of these 5 genes? This could be easily tested with luciferase reporter assays.

As stated above, when we re-analyzed our RNA seq we found an error in our differential gene analysis (two sample labels were switched during blinding). We corrected this error and re-ran the analysis, identifying a more robust set of overlapping genes (Fig. 5L, Supplemental Fig.S8). We discovered 42 overlapping genes, only 3 of which were predicted *let-7* targets (Fig. 5L, Supplemental Fig.S8). Two out of three, *Hmga2* and *Igf2*, are upregulated in LIN28B^{Wt1} and *let-7* KO mouse models, while one, *Nuak2*, is down regulated.

Both *Hmga2* and *Igf2* are well known and characterized *let-7* targets that have been implicated in Wilms tumor (and therefore in kidney development). *Nuak2* is a predicted *let-7* target according to Target scan, but not validated and its role in kidney development has not been documented. In this study we focus on *Igf2* given a burgeoning number of studies showing the importance of increased *Igf2* levels in Wilms Tumor and kidney development. *IGF2* is a growth-promoting imprinted gene commonly upregulated in Beckwith-Wiedemann and Perlman overgrowth syndromes that have high susceptibility to Wilms tumor (Bharathavikru Gene and Development 2018).

7. The discrepancy between let-7 KO and Lin28b over-expressing mice could arise from the fact that Lin28 regulates many other microRNAs and gene expression events. This should be adequately discussed and referenced with the existing literature.

This has been addressed in the discussion with the following statement: “The discrepancy between *let-7* KO and LIN28B^{Wt1} mouse models could also arise from the fact that Lin28b regulates other gene expression events by binding to mRNAs and modulating their translation independently of *let-7* (21-25), as well as from the fact that only 3 out of 12 *let-7* family members were suppressed during kidney development in the *let-7* KO mouse model as opposed to all of them in the LIN28B^{Wt1} animals”.

Reviewer #3 (Remarks to the Author):

This manuscript by Yermalovich et al is a follow up study of an earlier paper published in 2014. Compared to the previous study, the authors demonstrated that Lin28b, but not Lin28a, is inversely correlated with let-7 microRNA expression during kidney development, suggesting its predominant role in this process. Through an extensive set of experiments using transgenic mouse models, they demonstrated that transient overexpression of LIN28B at E16.5 led to prolonged kidney development and enhanced kidney function. This effect can be recapitulated in let-7 KO. In contrast, depletion of LIN28B resulted in impaired kidney function. The authors convincingly showed that changes of let-7 and downstream targets upon perturbation of LIN28B. These include Igf2/H19, which was previously associated with Wilms tumorigenesis, although details were not investigated. This is an interesting paper that demonstrated the endogenous function of LIN28B during development, which is currently not clear in the field.

Minor points:

1. p4. This study did not directly compare the importance of Lin28a and Lin28b in terms of their function in development (except for analysis of gene expression). I would suggest the authors tone down their conclusion "... implicating Lin28b (and not Lin28a) as playing the predominant role in nephrogenesis.

In this paper we examine nephrogenesis during the phase when Lin28a is not expressed at either the mRNA or protein level (from E13.5 to P2). In addition, even at its peak expression, E12.5, *Lin28a* mRNA is significantly and dramatically lower compared to *Lin28b* mRNA. Finally, it's been documented that only *LIN28B* and not *LIN28A* is activated in human Wilms tumor. While we cannot comment on the role of Lin28a during the early phases of kidney formation from the intermediate mesoderm, all of the above does suggest that Lin28b plays the predominant role in nephrogenesis.

2. page 6, microRNA names should be double checked, e.g., pre-let-7b1, g1, i1 should be pre-let-7b, g, i.

Thank you. It is corrected.

3. page 9, Lin28fl/fl should be Lin28b fl/fl.

Thank you. It is corrected.

4. Fig. S8, LIN28Bwt1 3 appears to be outlier. The authors might identify a more robust set of targets by excluding the outlier?

Thank you very much for noticing the outlier. It was extremely helpful. As per your suggestion, we reanalyzed the RNA seq data and found an error during our

differential gene analysis, in which samples were mis-labeled during blinding by our bioinformatician. So, we corrected and re-ran the analysis, identifying a more robust set of overlapping genes (Fig. 5L, Supplemental Fig.S8).

REVIEWERS' COMMENTS:

Reviewer #1 (Remarks to the Author):

The authors have done an excellent job responding to my comments . It is interesting that IGF2 and H19 are regulated at the transcriptional level . The authors might wish to comment briefly on the implications of this mechanistically .

I strongly recommend publication of this interesting paper .

--

Reviewer #2 (Remarks to the Author):

I am happy with the revised version of the manuscript. I think it's a very interesting observation that proves important role of microRNAs and RNA-binding proteins in mammalian development.

--

Reviewer #3 (Remarks to the Author):

I have no further comments for the revised manuscript.

Reviewer #1 (Remarks to the Author):

The authors have done an excellent job responding to my comments. It is interesting that IGF2 and H19 are regulated at the transcriptional level. The authors might wish to comment briefly on the implications of this mechanistically. I strongly recommend publication of this interesting paper.

Thank you. We have now addressed this point in the discussion with the following statement: "Both overexpression of *LIN28B* and knockout of *let-7* results in an upregulation of *Igf2* and *H19* genes at the transcriptional level. While the locus of *H19/Igf2* has been shown to be co-regulated by *cis*-regulatory elements in a tissue-specific manner, recent studies have also indicated that the *H19* antisense RNA transcript can also *trans*-activate the *Igf2* promoter¹. We have demonstrated in our study the up-regulation of both spliced and unspliced *Igf2* and *H19* transcripts. This observation begs the question of whether this is a direct consequence of the ability of Lin28b or *let-7* to bind to the *H19* RNA antisense transcript knowing that both Lin28 and *let-7* are capable of binding to *Igf2* mRNA²"

--

Reviewer #2 (Remarks to the Author):

I am happy with the revised version of the manuscript. I think it's a very interesting observation that proves important role of microRNAs and RNA-binding proteins in mammalian development.

Thank you.

--

Reviewer #3 (Remarks to the Author):

I have no further comments for the revised manuscript.

Thank you.

Reference:

1. Tran, V.G., *et al.* H19 antisense RNA can up-regulate Igf2 transcription by activation of a novel promoter in mouse myoblasts. *PLoS One* **7**, e37923 (2012).
2. Poleskaya, A., *et al.* Lin-28 binds IGF-2 mRNA and participates in skeletal myogenesis by increasing translation efficiency. *Genes Dev* **21**, 1125-1138 (2007).